# Long-Term Antibacterial Efficacy of Cetylpyridinium Chloride-Montmorillonite Containing PMMA Resin Cement

**DOI:** 10.3390/nano13091495

**Published:** 2023-04-27

**Authors:** Kumiko Yoshihara, Noriyuki Nagaoka, Yoji Makita, Yasuhiro Yoshida, Bart Van Meerbeek

**Affiliations:** 1National Institute of Advanced Industrial Science and Technology (AIST), Health and Medical Research Institute, 2217-14 Hayashi-cho, Takamatsu 761-0395, Kagawa, Japan; 2Department of Pathology & Experimental Medicine, Graduate School of Medicine, Dentistry and Pharmaceutical Sciences, Okayama University, 2-5-1 Shikata-cho, Kita-ku, Okayama 700-8558, Okayama, Japan; 3Advanced Research Center for Oral and Craniofacial Science, Okayama University Dental School, 2-5-1 Shikata-cho, Kita-ku, Okayama 700-8558, Okayama, Japan; nagaoka@okayama-u.ac.jp; 4Department of Biomaterials and Bioengineering, Faculty of Dental Medicine, Hokkaido University, Kita 13, Nishi 7, Kita-ku, Sapporo 060-8586, Hokkaido, Japan; 5Department of Oral Health Sciences, BIOMAT & UZ Leuven (University Hospitals Leuven), Dentistry, KU Leuven (University of Leuven), Kapucijnenvoer 7, 3000 Leuven, Belgium

**Keywords:** biofilm, cetylpyridinium chloride, montmorillonite, PMMA resin cement, antibacterial agents, dentin, bond strength

## Abstract

Despite being able to adhesively restore teeth, adhesives and cement do not possess any anticariogenic protection potential, by which caries recurrence may still occur and reduce the clinical lifetime of adhesive restorations. Several antibacterial agents have been incorporated into dental adhesives and cement to render them anticariogenic. Due to an additional therapeutic effect, such materials are classified as ‘dental combination products’ with more strict market regulations. We incorporated cetylpyridinium chloride (CPC), often used for oral hygiene applications, into montmorillonite (CPC-Mont), the latter to serve as a carrier for controlled CPC release. CPC-Mont incorporated into tissue conditioner has been approved by the Pharmaceuticals and Medical Devices Agency (PmontMDA) in Japan. To produce a clinically effective dental cement with the antibacterial potential to prevent secondary caries, we incorporated CPC-Mont into PMMA resin cement. We measured the flexural strength, shear bond strength onto dentin, CPC release, and the biofilm-inhibition potential of the experimental CPC-Mont-containing PMMA cement. An 8 and 10 wt% CPC-Mont concentration revealed the antibacterial potential without reducing the mechanical properties of the PMMA cement.

## 1. Introduction

Secondary caries is caused by cariogenic bacteria adjacent to a restoration [1,2,3,4]. To prevent secondary caries, antibacterial agents have been incorporated into tooth-restorative materials [4]. Metals, such as silver, zinc, and copper, have been used to render dental materials with antibacterial potential [5]. The antibacterial effect of inorganic compounds, such as quaternary ammonium salts, has also been investigated [6]. Several antibacterial monomers have been synthesized and added to resin-based materials. However, only a few monomers, such as 12-methacryloyloxy dodecylpyridinium bromide (MDPB), were approved by the Food and Drug Administration (FDA) [7]. MDPB can co-polymerize through its methacrylate group with other monomers, enabling it to be built into the polymer matrix of resin-based materials. Several in-vitro studies have proven the antibacterial effect of MDPB-containing adhesives [8]. However, once polymerized, the antibacterial potential of an MDPB-containing adhesive is limited to contact inhibition. The effect of MDPB on the inhibition of secondary caries could so far not be proven clinically [9].

With this study, we aimed to develop a dental cement with a controlled release of an antibacterial agent. We incorporated the antibacterial compound cetylpyridinium chloride (CPC) within the montmorillonite (Mont) to enable a gradual CPC release [10]. CPC is a cationic quaternary ammonium compound used in some types of mouthwashes, kinds of toothpaste, and throat sprays [10]. CPC can be loaded into the layered structure of Mont [10]. Our previous study revealed a higher CPC release for a CPC-Mont-containing adhesive versus an adhesive containing solely CPC [10]. Adding CPC-Mont did not reduce the mechanical properties of the adhesive [10]. Furthermore, the CPC-Mont-containing adhesive was found to be rechargeable with CPC [10]. Using this CPC-Mont technology, a CPC-Mont-containing tissue conditioner (tissue conditioner CPC, J. Morita, Tokyo, Japan) for dentures was marketed in Japan. Recently, CPC-Mont was approved by the Pharmaceuticals and Medical Devices Agency (PMDA) in Japan. In the continuation of the developing CPC-Mont-containing dental materials, CPC-Mont-containing PMMA resin cement was explored in this study for its mechanical properties and antibacterial effects. The null hypotheses tested in this study were that adding CPC-Mont to PMMA cement (1) did inhibit biofilm formation and (2) did not affect the cement’s mechanical properties.

## 2. Materials and Methods

### 2.1. Specimen Preparation

CPC-Mont was prepared as described previously and illustrated in Figure 1A [10]. Briefly, Na-Montmorillonite (Mont) (Kunipia-F, Kunimine Industries, Tokyo, Japan) with a cation-ion-exchange capacity of 1.19 mmol/g was selected for this study. CPC was supplied by Wako Pure Chemical Industries (Osaka, Japan) in a special grade of hexadecyl pyridinium chloride (C_21_H_38_NCl) solution. All of the stock solutions were prepared in ultrapure water (RFD 240NA, Advantec, Tokyo, Japan) for the process of anion uptake. In total, 2 g of Na-Mont was dispersed in 200 mL of deionized water and then stirred for 1 h. A pre-dissolved stoichiometric amount of CPC, equivalent to four times the cation exchange capacities of Na-Mont cat ions, was added to the dispersion slowly. Then the dispersion was continuously stirred at 23 °C for 1 day. The solid product was separated by centrifugation, rinsed with deionized water, and then dried at 60 °C for 1 day. Finally, the sample was ground into a powder. The sample particle size produced by the separation between the standard sieves was less than 25 μm.

CPC-Mont was added in a concentration of 4, 6, 8, 10, and 15 wt% to the PMMA powder of the commercial cement Superbond (‘SB’; Sun Medical, Moriyama, Japan). Each CPC-Mont/SB powder was mixed with liquid as recommended by the manufacturer. The commercial SB cement without CPC-Mont (0% CPC-Mont/SB) served as the control (Table 1).

### 2.2. X-ray Diffraction (XRD)

The crystal phases of the CPC-Mont/SB powders were determined by an X-ray powder diffractometer (CuKα1 1.5406 Å, RINT2500, Rigaku, Tokyo, Japan), operated at 40 kV, with a 200 mA current, and a scanning rate of 0.02°/s in 2 θ/θ scans.

### 2.3. Scanning (Transmission) Electron Microscopy (SEM/STEM) of CPC-Mont/SB Cross-Sections

A silicone mold (diameter: 10 mm; thickness: 2 mm) was filled with cement upon mixing 10 wt% of CPC-Mont/SB with liquid according to the manufacturer’s instructions. Fully set cement disks were next cross-sectioned and polished using an argon-ion beam (SM-090101 Cross-Section Polisher, Jeol, Tokyo, Japan), followed by an osmium coating prior to the Feg-SEM examination (JSM-6701F, Jeol). Additionally, ultrathin (70–90 nm) cross-sections of the cement disk were cut with an ultramicrotome (Leica EM UC6, Leica, Vienna, Austria) to be examined by the STEM at 200 kV (JEM-2100F, Jeol).

### 2.4. SEM of CPC-Mont/SB Powder and Cement Disks

CPC-Mont/SB (0, 4, 6, 8, 10, and 15 wt%) powder, deposited on the SEM-specimen holders, and CPC-Mont/SB (0, 4, 6, 8, 10, and 15 wt%) cement plates, prepared as mentioned above and polished with colloidal silica-polishing slurry (Compol 80, Fujimi, Kiyosu, Aichi, Japan), were subsequently coated with a thin carbon layer (SC-701CT, Sanyu Electron, Tokyo, Japan) to be examined by Feg-SEM (JSM-6701F, Jeol) at 5 kV with an annular semiconductor detector.

### 2.5. CPC Release

CPC-Mont/SB (0, 4, 6, 8, 10, and 15 wt%) cement disks, prepared as mentioned above, were soaked in 2 mL distilled water at 37 °C for 14 days. The water was changed every day. The amount of CPC release was quantified with an ultraviolet-visible (UV–vis) spectrophotometer (U-2010 Hitachi, Tokyo, Japan). It was calculated based on the peak height from an arbitrary baseline of 258.5 nm, which was attributed to the pyridinium ring structure of the CPC [10]. Three independent samples per condition were tested (*n* = 3) and the reproducibility was ensured by triplicate UV spectrometry measurements. The CPC release for each cement formulation data was statistically compared by a two-way analysis of variance (ANOVA) and Tukey’s post hoc test (α < 0.05) employing SPSS software (version 25, IBM, Armonk, NY, USA).

### 2.6. Biofilm Inhibition

CPC-Mont/SB (0, 4, 6, 8, 10, and 15 wt%) cement disks, prepared as mentioned above, were next placed in 24 well disks (one specimen per well). A volume of 2 mL of a 5 × 10^5^ CFU/mL *Streptococcus mutans* suspension in the Brain Heart Infusion (BHI) broth (Becton, Dickinson and Company, Franklin Lakes, NJ, USA), including 1% sucrose, which was for biofilm formation, was added to each well. After incubating at 37 °C for 24 h, the medium was changed. After 2 months, the samples were washed with distilled water and subsequently fixed using a 2.5 wt% aqueous glutaraldehyde solution (Nacalai Tesque, Kyoto, Japan). The disks were then dried at room temperature. A thin layer of osmium (Neoc STB, Meiwafosis, Tokyo, Japan) was deposited on the surface of the disks. The biofilm that was formed on the disks was then observed using Feg-SEM (JSM-6701F, Jeol). This measurement was conducted in triplicate (*n* = 3 sites on 3 disks).

### 2.7. Flexural Strength

To measure flexural strength (*n* = 10 per group), Teflon molds (25 mm length × 2 mm width × 2 mm height) were filled with the CPC-Mont/SB cement formulations. The specimens were kept to set for 30 min, upon which they were removed from the Teflon molds and immersed in distilled water. In total, 30 specimens per CPC-Mont/SB cement formulation were prepared. Half of these specimens were tested immediately, while the other half were stored in distilled water for 3 weeks. The specimens were then tested in a three-point bending test, with a 20 mm span and a load speed of 0.5 mm/min (Model 5565, Instron, Canton, MA, USA), according to the ISO 9917-2 (1996) standard. The flexural strength was also determined from the highest stress measured when the specimens failed. A statistical analysis was performed with a two-way ANOVA and Tukey’s honestly significant difference (HSD) test at a significance level of α = 0.05 using SPSS (IBM).

### 2.8. Shear Bond Strength to Dentin

The occlusal third of molars crowns, extracted for dental reasons upon approval by the Commission for Medical Ethics of Okayama University (under file number #1606-020), was removed at the level of mid-coronal dentin, upon which the sectioned teeth were embedded in epoxy resin (EpoFix, Struers, Ballerup, Denmark). The exposed dentin was ground using 600-grit Sic paper (WTCC-S, Nihon Kenshi, Fukuyama, Japan) to prepare a standard smear layer. A Teflon mold with a cylindrical hole (diameter: 3.6 mm, height: 2 mm) was clamped onto the dentin surfaces and filled with each cement formulation. Thirty specimens were prepared per experimental group. Ten specimens were subjected to a shear-bond-strength testing protocol after storage for 24 h in water at 37 °C, while the other two 10-specimen sets were stored in water at 37 °C for 2 and 6 months, respectively. The specimens were stressed at a cross-head speed of 1 mm/1 min until failure using a universal testing device (AGS-X, Shimadzu, Kyoto, Japan). For statistical comparison, a two-way ANOVA followed by Tukey’s post hoc tests (α < 0.05) were employed with *p* < 0.05 being considered statistically significant.

### 2.9. Biofilm Formation on the Fractured Surface of Shear-Bond-Strength Specimens

The dentin parts of the fractured 6-month shear-bond-strength specimens were placed in 24 well disks. A volume of 2 mL 5.0 × 10^5^ CFU/mL *S. mutans* suspension in BHI, containing 1% sucrose, was added to each well. After incubation at 37 °C for 24 h, the specimens were washed using distilled water and then fixed with 2.5 wt% glutaraldehyde (Nacalai Tesque, Kyoto, Japan). Subsequently, a thin layer of osmium was deposited on their surfaces, upon which the biofilm formed on the disks was observed using Feg-SEM (JSM-6701F, Jeol). These measurements were conducted in triplicate (*n* = 3 sites on three disks).

## 3. Results

### 3.1. XRD 

XRD is an important method to characterize the structure of crystalline material. The XRD of SB revealed one broad peak at 2θ = 13.06° (0.297 nm). The mont revealed two weak peaks at 2θ = 7.14° (0.12 nm), while the CPC-Mont and 10 wt% CPC-Mont/SB revealed several identical peaks at 2θ = 2.14° (4.22 nm), 2θ = 4.28° (2.08 nm), 2θ = 6.33° (1.40 nm), and 2θ = 8.52° (1.04 nm) (Figure 1B).

### 3.2. SEM and STEM of CPC-Mont/SB Cement-Disk Cross-Sections

The SEM of the 10 wt% CPC-Mont/SB cement-disk cross-sections revealed irregular fibrous structures with an approximate size of 30 µm (Figure 1C(a,b)). Low-magnification STEM imaging of the ultrathin 10 wt% CPC-Mont/SB cement-disk cross-sections revealed 2 µm filler-like structures (Figure 1C(c)). High-magnification STEM imaging disclosed a layered structure with a 2 nm layer thickness (Figure 1C(d)).

### 3.3. SEM of CPC-Mont/SB Powder and Disks 

The SEM of The CPC-Mont/SB powder revealed various sizes of irregularly shaped PMMA polymers. Among these polymers, irregularly shaped filler particles with a bright-gray color were observed (Figure 2A: arrows), which increased in number with higher CPC-Mont concentrations.

The SEM of cross-sectioned hardened PMMA cement disks revealed clearly differently shaped, gray-colored structures that increased in number with a higher CPC-Mont/SB concentration (Figure 2B).

### 3.4. CPC Release 

The UV-spectroscopy revealed that the higher CPC-Mont/SB-concentrated cement disks released more CPC (Figure 3). The amount of CPC release gradually decreased as time progressed. After 14 days, all specimens were still releasing CPC.

### 3.5. Biofilm Inhibition 

The surface of each specimen was evaluated by SEM for biofilm formation. (Figure 4) The wow-magnification of the 8, 10, and 15 wt% CPC-Mont/SB cement surfaces revealed scratches. This means the cement surfaces were exposed. As a result, no biofilm was formed on the 10 and 15 wt% CPC-Mont/SB cement surfaces that were exposed for 2 months to the *S. mutans* medium, which was changed every 24 h. Some biofilm clusters were observed on the 8 wt% CPC-Mont/SB cement disks, while the 4 and 6 wt% CPC-Mont/SB disks did not inhibit biofilm formation. The SB control (without the CPC-Mont/SB) revealed biofilm formation with cracks.

### 3.6. Flexural Strength 

Figure 5 showed the result of the flexural strength test. No significant difference in the flexural strength was recorded among all the experimental formulations, except for the 15 wt% CPC-Mont/SB, which yielded a significantly lower flexural strength. Water storage for 2 and 6 months did not significantly decrease flexural strength.

### 3.7. Shear Bond Strength to Dentin 

Figure 6 shows the results of the shear bond strength test. There were no significant differences in the 24 h μTBS, which was immediately tested after the specimen’s preparation to dentin was recorded among all the experimental formulations, except for the significantly lower μTBS recorded for the 15 wt% CPC-Mont/SB. While the shear bond strength of all experimental CPC-Mont/SB formulations decreased upon 2 and 6 month storage, this decrease was not different from that of the control cement (0 wt% CPC-Mont/SB), with the 15 wt% CPC-Mont/SB, again, revealing the significantly lowest μTBS upon the 2 and 6 month water-storage aging.

### 3.8. Biofilm Formation on the Fractured Surface of Shear-Bond-Strength Specimens

To evaluate the effect of secondary caries inhibition in the debonding area, the biofilm inhibition ability on the dentin side of the fracture surface was tested (Figure 7). A low magnification of the 6, 8, 10, and 15 wt% CPC-Mont/SB specimens revealed scratches, which was the dentin surface. This means that biofilm inhibition was observed on the fractured shear-bond strength of the 6, 8, 10, and 15 wt% CPC-Mont/SB specimens after 6 months of storage. Some biofilm formation was observed on 4-wt% CPC-Mont/SB specimens. A high magnification image of the 0 wt% CPC-Mont/SB revealed the existence of the *S. mutans*. Additionally, the biofilm was completely covered for the 0 wt% CPC-Mont/SB. A high magnification image of the 0 wt% CPC-Mont/SB revealed that the *S. mutans* had completely covered the surface. This means that no biofilm inhibition was recorded for the 0 wt% CPC-Mont/SB.

## 4. Discussion

This study aimed to investigate the antibacterial efficacy of the novel inorganic antibacterial agent CPC-Mont when contained in a PMMA resin cement. We demonstrated a distinct CPC release from the cement with a higher CPC release recorded from the higher CPC-Mont-concentrated cement formulations. The 10 wt% CPC-Mont/SB cement formulation inhibited the biofilm formation without losing mechanical properties.

Montmorillonite (Mont) is known as a clay mineral that has substantial isomorphous substitutions. The exchangeable cations of the 2:1 layer of Mont are balanced by the negative charge resulting from the isomorphous substitution [11,12]. Montmorillonite can adsorb long-chain quaternary alkylammonium ions when their amounts increase the cation-exchange capacity by up to 2–4 times to form organo-montmorillonite. The latter acts as an adsorbent for the uptake of anions from the aqueous solutions [11]. As shown by XRD, the layer spacing of the CPC-Mont was larger than that of Mont, confirming that CPC was inserted in the montmorillonite [10]. In a previous study, when the CPC-Mont was incorporated into a dental adhesive, CPC was released for a longer time than when the adhesive was loaded with solely CPC [10]. Furthermore, the CPC-Mont-containing adhesive revealed higher mechanical properties than the adhesive loaded solely with CPC [10].

The SEM of the CPC-Mont/SB cement formulations revealed a 30 µ island-like CPC-Mont structure in the resin matrix. The STEM confirmed a 2 nm layered structure. Higher CPC-Mont-concentrated cement formulations exhibited more CPC-Mont filler particles in a concentration dependency, as imaged by SEM [10]. The amount of CPC release also depended on the initial CPC-Mont amount added to the SB, with higher-concentrated CPC-Mont/SB cement formulations releasing more CPC. This finding was in agreement with a previous study that demonstrated a higher CPC release from a 3 wt% CPC-Mont-loaded adhesive than from the lower concentrated CPC-Mont adhesive formulations [10]. In general, higher concentrated CPC-containing materials release more CPC. Namba et al. (2009) reported that a 3 wt% CPC-containing composite resin released more CPC than a 1 wt% CPC-containing formulation [13]. Al-musallam et al. (2006) tested 2.5, 5, and 10 wt% CPC-containing adhesives and confirmed a higher CPC release by the higher CPC-concentrated formulations [14]. In this study, the CPC release detected by the UV-spectroscopy was found to decrease as time progressed.

We also tested the antibacterial efficacy of the CPC-releasing cement formulations. Several test methods can be used to measure antibacterial efficacy [15,16]. Often used is the agar-diffusion test, disks containing antibiotics are placed on an agar disk inoculated with bacteria, upon which the disk is kept to incubate. If the antibacterial components kill the bacteria or stop the growth of the bacteria, the areas around the wafer where the bacteria are not allowed to grow will be clearly apparent. This area is referred to as the zone of inhibition [6]. Although this protocol is simple, the test method has many limitations, as the outcome also depends on diffusion and the physical properties of the tested materials [17]. Nevertheless, several studies used this agar-diffusion test to measure long-term antibacterial effects [18,19,20]. However, this method does not consider the long-term antibiotic release by diffusion from the materials. Furthermore, dental materials with antibiofilm capabilities are desired as follows: (1) to inhibit the initial binding of bacteria, (2) to prevent biofilm growth, (3) to affect the microbial metabolism in the biofilm, (4) to kill the biofilm of the bacteria, and (5) to detach biofilms [21]. More specifically, biofilm formation may lead to the invasion of the bacteria along the restoration-tooth interface and, in time, to secondary caries. Agar diffusion tests can demonstrate the growth inhibition of the micro-organisms. To observe the bactericidal and microbicidal activity, direct contact tests are used [6]. This method involves fluorescence microscopy, confocal laser scanning microscopy (CSLM), and SEM to image capture the biofilms growing on material disks [6,21,22,23,24,25]. In this study, we measured the CPC release and investigated the biofilm inhibition by SEM using the same broth amounts.

The SEM of the 2-month biofilm formation revealed that the higher concentrated CPC-Mont cement formulations led to more prominent biofilm-inhibition effects. As a control, CPC-Mont free, with the commercial dental PMMA cement Superbond (0 wt% CPC-Mont/SB), was tested. The surface of the 8, 10, and 15 wt% CPC-Mont/SB specimens still revealed scratches, indicating that they were not covered by biofilm. Some biofilm clusters were observed for the 8 wt% CPC-Mont/SB specimens. The higher magnifications of the 10 and 15 wt% CPC-Mont/SB specimens, however, did not show any microorganisms on the surface. By loading the CPC into the Monte by ion exchange, the CPC was, relatively, slowly released from the CPC-Mont/SB cement formulations, by which the antibacterial effects were still detected after 2 months [10]. The slow CPC release should be attributed to the specific layered and charged structure of the CPC-Mont. These findings indicate that the first null hypothesis, that adding CPC-Mont to PMMA cement did inhibit the biofilm formation, was accepted when the CPC-Mont was added in a sufficiently high concentration of at least 10 wt%.

The mechanical properties and bonding ability are primordial for adhesive luting agents. In this study, the mechanical properties were measured in terms of flexural strength. When the CPC-Mont, in a concentration of up to 10 wt%, was added to the PMMA resin cement Superbond (SB; Sun Medical), its immediate flexural strength only slightly decreased, but not significantly. After 6 months, the strength of the 0 wt% CPC-Mont/SB, with the commercial dental cement Superbond, decreased, while the samples with the 4–10% addition showed no decrease in strength. On the other hand, the sample with the 15% addition had a lower immediate strength and lower durability. The flexural strength of the 15 wt% CPC-Mont/SB cement formulation was significantly lower, by which the second null hypothesis, that adding the CPC-Mont to PMMA cement did not affect the cement’s mechanical properties, was accepted when the CPC-Mont was added up to 10 wt%. Polymer-clay interactions have been studied before [26]. Adding a maximum of 7% Montmorillonite-polyamide 6 improved the mechanical properties of its polymer and also improved the thermal resistance and barrier effect [27]. Mont can be incorporated in several polymers, such as poly(ethylene oxide), poly(vinyl alcohol), polyamides, and epoxy polymer resins [26]. Montmorillonite (Mont) not only exists abundantly in nature, but also has a crystal grain size with disk-like nanoparticles in a high aspect ratio (the ratio of its sizes in different dimensions) [26]. Therefore, by adding 1% Mont to a polymer, the barrier property against moisture, solvent, the steam of chemicals, and oxygen is improved [28]. Mont-containing PMMA was also investigated [26]. Compared with neat PMMA, PMMA/Mont nanocomposites exhibited several favorable properties, such as enhanced mechanical properties, higher glass-transition temperature, better thermal stability, fire retardancy, and improved solvent resistance and anticorrosion ability [26]. The incorporation of all types of Mont increased the flexural modulus and strength, remarkably owing to the high stiffness and aspect ratio of silicate [26,27]. More specifically, the organo-modified montmorillonite improved the mechanical properties [29]. Although the 10 wt% CPC-Mont revealed stable mechanical properties, the 15 wt% added material has reduced strength in this study, which suggests that the optimum concentration of addition should be considered.

PMMA resin could penetrate the fibrous structure of the CPC-Mont, as evidenced by the STEM, and may improve the mechanical properties. Furthermore, SB contains the acidic functional monomer, which is 4-Methacryloxyethyl trimellitate anhydride (4-META) that can bond to tooth structure [30]. The 4-META/MMA-tri-*n*-butylborane (TBB) resin formulation of the SB was shown to durably bond to enamel and dentin, especially to deep and cervical dentin, this, in part, owing to its high degree of polymerization in a wet dentin environment compared with other resin-based cements [31]. However, Superbond and Superbond C&B (Sun Medical) were documented with a bond strength that decreased upon 6 months of water-storage aging [32]. Additionally, the result of the shear bond strength of the control cement SB, to which no CPC-Mont was added (0 wt%), slightly decreased after 2 and 6 months of water storage. Otherwise, the CPC-Mont/SB cements, with a CPC-Mont concentration of up to 10 wt%, revealed a similar or even higher bond strength to dentin, which potentially must be ascribed to the increasing flexural strength. However, the higher 15 wt% CPC-Mont concentration decreased both the flexural strength and shear bond strength to dentin. While some CPC-Mont amounts improved the mechanical properties, hereby probably acting as a filler, a higher CPC-Mont amount may decrease the flexural strength due to the de-bonding between the CPC-Mont and PMMA resin in the absence of silane coupling and/or due to the exfoliation of the Mont layers [33].

Finally, we evaluated biofilm inhibition at the dentin surface of the specimens that were fractured during the shear bond-strength testing after 6 months of water storage. The 0 wt% CPC-Mont/SB control specimens did not inhibit the biofilm formation, while the 4 wt% CPC-Mont/SB specimens slightly inhibited the biofilm formation. The CPC-Mont/SB specimens with a CPC-Mont/SB concentration higher than 6 wt% inhibited the biofilm formation. This indicates that the latter CPC-Mont/SB cement formulations still exhibited the antibacterial effects at the cement-dentin interface of the specimens, which were immersed in distilled water for 6 months. This indicates that the antimicrobial properties were maintained at the cement-dentin interface. Secondary caries originates in the presence of bacterial biofilms, most likely when associated with a discontinuity or gap at the margin of the restoration with the tooth structure. Such a marginal gap may result from an initially defective margin or from interfacial degradation [34]. Secondary wall lesions were reported to develop at 30 µm gaps [35]. Materials releasing CPC-Mont, such as the CPC-Mont-containing PMMA resin cement investigated in this study, demonstrated the potential in this study to inhibit bacterial penetration, possibly even when a gap is formed.

## 5. Conclusions

In this in-vitro study, a PMMA resin cement loaded with CPC-Mont was shown to gradually release CPC and inhibit biofilm formation up to 2 months after placement. Furthermore, the resin cement-dentin interface still showed antibacterial effects following water storage for 6 months. Evidence is now needed to confirm the potential of the CPC-Mont-containing PMMA resin cement to clinically inhibit secondary caries. More evidence is needed for more long-term efficacy and the clinical effects of the inhibition of secondary caries.

## Figures and Tables

**Figure 1 nanomaterials-13-01495-f001:**
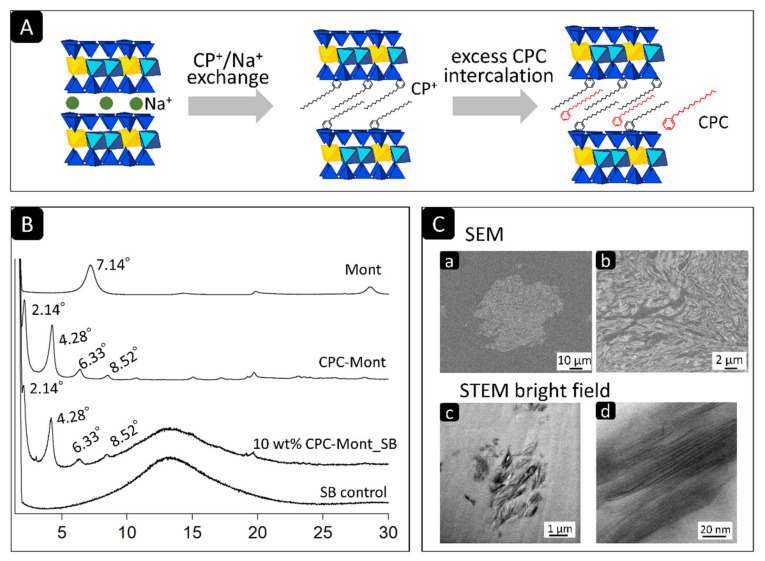
(**A**) Schematic illustrating the synthesis of CPC-Mont. Cetylpyridinium ions (CP^+^) are ion-exchanged with sodium ion (Na^+^) of montmorillonite and CPC are loaded on montmorillonite. (**B**) X-ray diffraction (XRD) patterns of Montmorillonite (Mont) powder, CPC-containing Montmorillonite (CPC-Mont) powder, 10 wt% CPC-Mont/SB powder, and Superbond (Sun Medical) PMMA cement powder (SB control). (**C**) Low- and high-magnification SEM photomicrographs in (**C**) (**a**,**b**), illustrating an about 30 µm gray-colored irregular and fibrous structure, representing CPC-Mont. Low- and high-magnification STEM photomicrographs in (**C**) (**c**,**d**), showing a 2 µm filler-like layered structure with 2 nm thick layers.

**Figure 2 nanomaterials-13-01495-f002:**
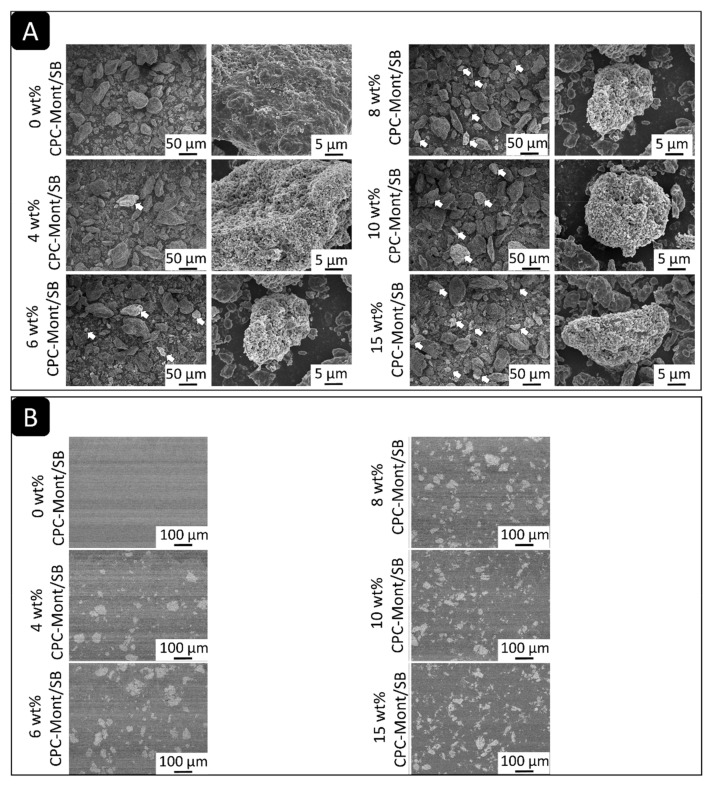
(**A**) The SEM photomicrographs of the CPC-Mont/SB powder formulations revealed variously sized particles with irregular shapes. Among these particles, bright-gray colored irregularly shaped filler particles were observed (arrows). Higher CPC-Mont concentration generated more gray-colored filler particles. (**B**) The SEM photomicrographs of the cross-sectioned hardened PMMA cement disks clearly disclosed the grey and irregularly shaped filler particles when CPC-Mont was added.

**Figure 3 nanomaterials-13-01495-f003:**
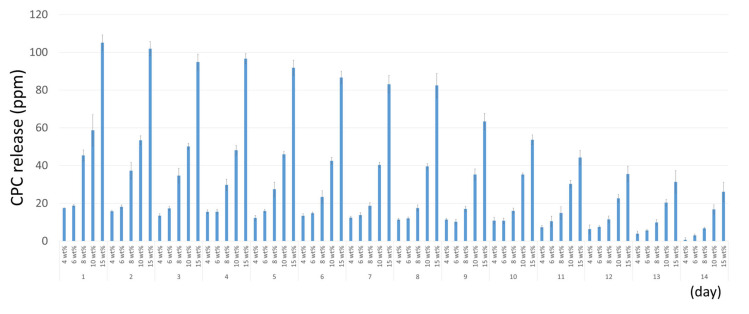
Two-week CPC release from the CPC-Mont/SB cement measured by UV spectroscopy.

**Figure 4 nanomaterials-13-01495-f004:**
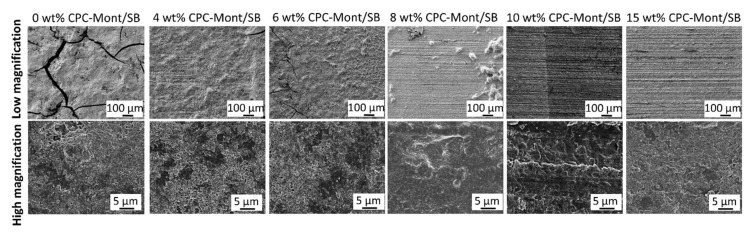
*Streptococcus mutans* biofilm formation on the CPC-Mont/SB cement surfaces, after having been exposed for 2 months to *S. mutans* medium, which was changed every 24 h.

**Figure 5 nanomaterials-13-01495-f005:**
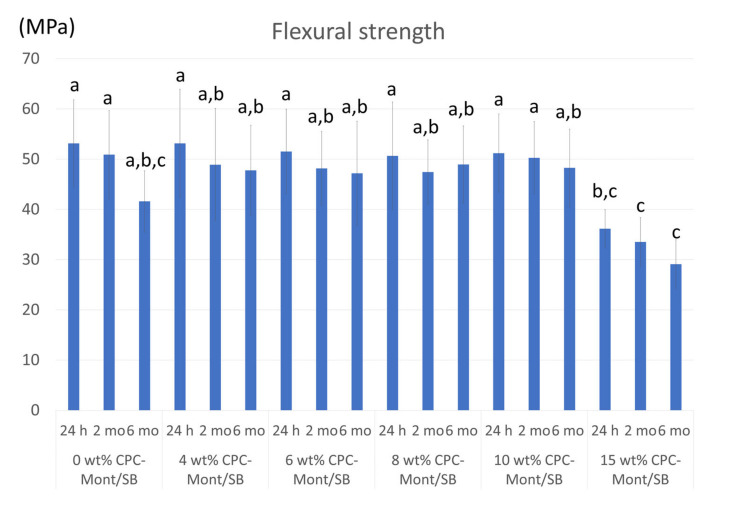
Flexural strength after water storage for 24 h, 2, and 6 months. The same letter in the same column means no significant difference (*p* > 0.05).

**Figure 6 nanomaterials-13-01495-f006:**
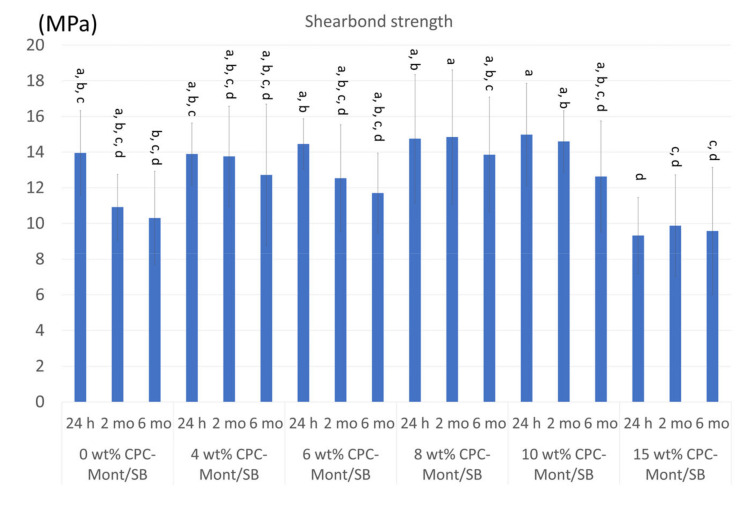
Shear bond strength onto dentin after water storage for 24 h, 2, and 6 months. The same letter in the same column means no significant difference (*p* > 0.05).

**Figure 7 nanomaterials-13-01495-f007:**
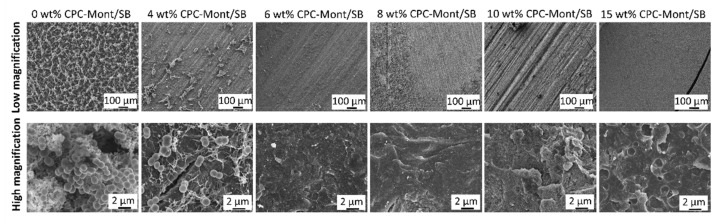
Biofilm formation on fractured dentin-side surfaces after shear bond-strength testing, as imaged by SEM.

**Table 1 nanomaterials-13-01495-t001:** Composition of the CPC-Mont/SB experimental formulations.

	(wt%)	0	4	6	8	10	15
POWDER (wt%)	Superbond (SB)	100	96	94	92	90	85
	CPC-Mont	0	4	6	8	10	15
LIQUID	MMA, 4-META	4 monomer drops mixed with 1 catalyst drop, and 0.2 mL powder					
	TBB catalyst						

## Data Availability

The data presented in this study are available from the corresponding author, K.Y., upon reasonable request.

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
