# Peer review of "Long-Term Antibacterial Efficacy of Cetylpyridinium Chloride-Montmorillonite Containing PMMA Resin Cement"

_nanomaterials, 2023, doi:10.3390/nano13091495_

Round 1

Reviewer 1 Report

In the manuscript, the authors developed an antibacterial dental cement based on cetylpyridinium chloride trapped in Montmorillonite (CPC-Mont) containing PMMA. The work is interesting, and the antibacterial ability of the material is promising, but some issues should be addressed before I can recommend its publication:

  1. The manuscript should be carefully reviewed for style and grammar, as text comprehension results are sometimes difficoult.
  2. I suggest merging the results and discussion sections to simplify the reading and make figures easier to consult. 
  3. The composition of the samples is not clear; in materials and method, it seems that they contain the same amount of CPC-Mont and growing concentrations of PMMA, while in the discussion the concentration of CPC-Mont is considered the variable parameter. Also, on page 6 lines 242-243, the sentence “The amount of CPC-release also depended on the initial amount of SB loaded in CPC- Mont” seems to suggest a further parameter to be considered. Please clarify, possibly including a schematic (or adding subsequent steps in fig.1A)
  4. How was the protocol for the biofilm formation test selected? It would be helpful to add a reference.
  5. Page 4, Lines 174-179, and page 6 lines 237-240: Which figure does the description refer to? It would seem to concern fig.1a that, however, is described in the caption as relating to the CPC-Mont samples. If the paragraphs describe figure 1a, is it possible to specify why the clearest island visible to the SEM is defined as a structure? The scale bar is not readable in the figure however, the dimensions appear to be closer to 30 μm, as in the discussion section than 20 μm as indicated in the results section. Please check and solve inconsistencies. 
  6. Pag. 7 line 302: the authors state that the results obtained are in line with those of the literature, but the addition of Mont in their case leads only to a slight improvement of the mechanical properties (or worsening in the case of 15%) against the remarkable effect found in other works. Could you elaborate on the comment?
  7. Indicate the figures numbers in the text in the most appropriate position. Only Fig1a. is properly mentioned. Page 5 line 208: substitute Fig. XX with the proper label.
  8. Figure 1b: the spectrum of Na-Montmorillonite is not present.
  9. Figure 3: did the author consider representing the CPC release as cumulative release vs time to possibly examine some kinetic aspect? In any case, a title for the y-axis, the CPC concentrations at time zero, and colors to identify the different samples would help the legibility of the graph. 

Author Response

Thank you very much for your comments. I prepared response letter and I also totally revised the manuscript.

Reviewer 2 Report

The manuscript I read wasn't proof-read for the English and it is in many parts unintelligible and rich of misspellings. Particularly, the discussion clearly wasn't  checked for clarity and for coherence.  Figures are never cited/recalled in the text and it is difficult to follow the results and discussion session without those references.

Bibliography isn't correct: numbers in the text do not refer to the correct publication, many references refer to more than one publication and the style/format is not uniform.

 Besides, figures are low quality and not unequivocally indexed.

 I suggest that authors restart the submission with a more advanced manuscript.

 More in details:

Materials

 Biofilm formation. A control for contamination after 2 months incubation has to be clearly described.

 Results

 XRD analysis. Explain the results of this technique and the graph reported in Figure 1B.

 Figure 4, the second row is not numbered neither described.

 Figure 5 and 6. A, b, c indexing has no legend.

 All data reported in the results have to report the figure number. All figures have to be clearly described in the text not only for what they show but also for their meaning and interpretation, see figure 1. Composed figures have to be numbered univocally.

 All abbreviations have to be written in full at least the first time (ex, line 206 page 5).

 Figure xx wasn’t numbered.

 Discussion.

 This part has to be rewritten completely and proof-read by professionals. The significance of the results has to become clear.

 Give prominence to your study compared to those that you cite, especially when citations date back more than 10 years.

 Check all spellings and repetitions.

 Some unclear wordings, for ex “4-MET” page 7, “high aspect”.

 Give the benchmark of the characteristics of your new materials, for example shear bond strength, what requirement do you need to hit?

Bibliography has to be thoroughly rechecked.

Author Response

Thank you very much for your comments. We made response letter and we totally rivised the manuscript. 

Reviewer 3 Report

Authors conducted the in vitro study on the CPC-Mont with PMMA resin in the case at which biofilm formed. This paper was well designed and wrote, which can be accepted after some minor revisions.

1.      Abstract: the lengthy abstract should be reduced, because some information should appear in the introduction section. some specific data should be discussed in the abstract section.

2.      Introduction: some more recent work should be comparatively discussed in the introduction section to reveal the novelty of this work.

3.      Figures: the scale bars in all SEM images are hard to see and should be re-marked. The curves in Fig. 1b should be re-plotted to make them clear.

Author Response

Thank you very much for your comments. We made a response letter and totally revised the manuscript.

Reviewer 4 Report

Abstract

The abstract must be corrected, after considering the corrections proposed in the article, namely in the conclusions.

Introduction

Line 43 – reference is missing

Line 59 – reference is missing

Line 50 to 64 – there aren’t any reference. This “previous study” was published”? If not, you can’t refer the study.

The null hypothesis is missing.

Materials and Methods

“…(Fig. xx) …” – The authors must refer the number of the figure

The figures must be referred to throughout the text.

How many groups were constituted, what is the number of groups (name of groups). Did the authors not do a control group and/or a negative control group? This is not clear…

Results

The figures must be referred to throughout the text.

Discussion

The reference to the figures must appear in the materials and methods and in the results - they must not be present in the discussion.

Lines 235; 240; 245 – references are missing.

“…In our previous study,…” - Authors should substitute "previous studies...".

“…This  agreed with our previous study. CPC-Mont specimens at concentrations equal…” - Authors should substitute "previous studies...".

“…Al-musallam tested 2.5, 5, and…” – Al-musallam et al.

“…If an antibiotic stops the bacteria from grow-258 ing or kills the bacteria, there will be an area around the wafer in which the bacteria 259 have not grown adequately to be visible. This is referred to as the zone of inhibition. 260 6 This protocol was simple. It had many limitations as it is dependent on diffusion 261 and the physical properties of the tested materials. 17 Conversely, several studies 262 used this agar diffusion test for long-term antibacterial effects. 18–20 However, this 263 method did not consider antibiotic losses owing to diffused from materials in the 264 long term…” - Authors should rewrite this part of the text to better understand what they want to say or justify

Conclusions

Authors should refer to the need for long-term studies, not only to study sustained release as well as structural changes and mechanical properties of the adhesive at the interface.

References

Authors should format references

Author Response

Thank you for your comments. We made a response letter and revised the manuscript.

Reviewer 5 Report

It is a very interesting research regarding the cetylpyridinium chloride long time release from a PMMA resin cement containing Montmorillonite to assure long term antibacterial effect. It is a wise approach due to Montmorillonite ability to store liquid solutions inside of its layered nano and microstructure and to release it progressively in time. Congratulation for the proper adaptation of XRD working regime, the speed of 0.02 deg/second allows better development of the low angle peaks of phyllosilicates such Montmorillonite. The obtained results seems to be good but are presented in a very poor manner that requires proper revision, especially the microscopy images are of low quality and do not allow to follow the features described in text. The reference citation in text and the reference list presentation are poor and require improvements. The manuscript must be revised according to the comments detailed below:

Comment 1) All microscopy images are of very poor quality and do not allow observing of discussed features and their sizes are too small. Their scale bars are completely unreadable; therefore it is doubtful that discussed nanostructural details are actually present in those images. Text beside SEM images is also unreadable. Please revise all microscopy images to assure a proper quality, size (allowing a good visibility of the micro and nanostructural details) and a good visibility and readability of the scale bar and text beside.

Comment 2) Figure 1b – XRD pattern for CPC-Mont is missing. Figure 1b must be revised by adding the missing XRD pattern.

Comment 3) Statistical analysis aspects regarding flexural strength and shear bond strength are poorly presented and discussed, consider their improvement.

Comment 4) ,,Scanning Transmission Electron Microscopy (STEM) observation” – the used device name and description is missing, it must be added.

,,Shear-bond strength” – the used testing machine name and description is missing, it must be added.

Comment 5) Figures are not mentioned in text at results section nor at discussion. According to the Nanomaterials template figures must be placed in text as close to their first mention. Therefore, all figures must be removed from the end of the manuscript and placed at their proper position at results section.

Comment 6) References citation in text is very poor and it is not clear to which phrase belongs. All references must be cited in text according to the Nanomaterials template and related instructions. They are numbers in square brackets placed inside the phrase or at the end of the phrase before the final point as example [22]. Therefore, reference citation in whole manuscript must be revised.

Comment 7) References [31 – 33] are missing from the list. They must be added to the list or removed from the text.

Comment 8) Reference list is not presented according to the template requirements, consider revision.

Author Response

(The authors gave the same response as above.)

Round 2

Reviewer 2 Report

Authors have improved the manuscript in this amended version. Still, the study has flaws to be corrected before publishing.

The English has to be revised by professional reviewing, for example verb tenses have to be uniformly adjusted all over the document.

Numbers of figures should be removed from titles, they have to be only in the text.

Reviewer 4 Report

Conclusions: The authors must confirm or refuse the null hypothesis

Reviewer 5 Report

All requested correction and completions were well effectuated to the manuscript.
